# Vascularization Reconstruction Strategies in Craniofacial Bone Regeneration

Jiping Chen [ID], Yu Da, Jing Yang, Guirong Zhu and Haiyan Qin *

Department of Stomatology, Drum Tower Hospital, The Affiliated Hospital of Nanjing University Medical School, Nanjing 210000, China; chenjiping@smail.nju.edu.cn (J.C.); rain_da@126.com (Y.D.); jinyang19961104@163.com (J.Y.); 15250968551@163.com (G.Z.)
* Correspondence: haiyanandrew@163.com

**Abstract:** Craniofacial bone defects are usually secondary to accident trauma, resection of tumor, sever inflammation, and congenital disease. The defects of craniofacial bones impact esthetic appearance and functionality such as mastication, pronunciation, and facial features. During the craniofacial bone regeneration process, different osteogenic cells are introduced, including primary osteoblasts or pluripotent stem cells. However, the defect area is initially avascular, resulting in the death of the introduced cells and failed regeneration. Thus, it is vital to establish vascularization strategies to build a timely and abundant blood vessel supply network. This review paper therefore focuses on the reconstruction of both osteogenesis and vasculogenesis. The current challenges, various strategies, and latest efforts applied to enhance vascularization in craniofacial bone regeneration are discussed. These involve the application of angiogenic growth factors and cell-based vascularization strategies. In addition, surface morphology, porous characters, and the angiogenic release property of scaffolds also have a fundamental effect on vasculogenesis via cell behavior and are further discussed.

**Keywords:** craniofacial bone; osteogenesis; vasculogenesis; bone tissue engineering; blood vessel; angiogenesis growth factor; biocompatible materials

## 1. Introduction

Craniofacial bone provides support for adjacent craniofacial soft tissues (especially the attachments of mastication-related muscles) and anchorage for dental structures [1]. The defects of craniofacial bones, secondary to accident trauma, congenital disease, tumor resection, and inflammation [2–5], impact esthetic appearance and functionality of the craniofacial complex, such as mastication, pronunciation, and facial features. Furthermore, craniofacial bone is highly vascularized, and its functions depend a lot on an unobstructed and well-organized vascular network. With the intact vessels, sufficient oxygen and nutrients can be supplied, guaranteeing the proliferation and viability of cells [6]. At the same time, the metabolic waste of cells can be taken away [7]. Therefore, it is well-recognized that the prompting of vasculogenesis is beneficial for reinforced bone functions. After the craniofacial bone defect occurs, osteogenic cells such as primary osteoblasts or pluripotent stem cells are recruited in order to generate neobones. However, the defect area is initially avascular, resulting in the death of the recruited cells and failed regeneration [6]. Thus, it is vital to establish vascularization strategies to build a timely and abundant blood vessel supply network [8].

Clinically, the damaged craniofacial bone can be reconstructed with a series of surgical operations. More than 90% of grafts used are autologous or allogenic transplantations, which are recognized as the "gold standards" [9]. However, challenges, such as donor site morbidity, pain, infection, and additional economic burden, are still unmet [10]. More importantly, vasculogenesis in the depth of the defect also cannot be well-established, which leads to the necrosis of transplanted grafts [11–13].

Rapid developments in bone tissue engineering bring new hope for solving the urgent problems and provide more strategies for neovascular networks and craniofacial bone tissue regeneration. During the past decades, numerous studies have been accomplished, introducing different angiogenic cells and growth factors based on biocompatible scaffolds for rebuilding vessel networks in craniofacial bone defects [9,14,15]. This work, presenting a first-time comprehensive review of recent advances of vascularization strategies in craniofacial bone tissue regeneration, overviews the current challenges, various strategies, and the latest efforts applied to enhancing vascularization in craniofacial bone regeneration.

## 2. Challenges of Vascularization in Craniofacial Bone Regeneration

Vasculogenesis and angiogenesis are two well-known approaches by which embryonic blood vessels develop. Vasculogenesis means that new blood vessels are formed in suit by endothelial progenitor cells and then coalesce with elongating vessels. In contrast, angiogenesis, assumed as the more prevalent way of vascularization, is related to new capillaries by budding, branching, and elongation of existing vessels [16–18]. While the specific angiogenesis mechanism during craniofacial bone defect regeneration remains unexplicit, some inspirations can be obtained from the craniofacial bone formation. Unlike the endochondral ossification pattern of the appendicular skeleton, most craniofacial bones display an intramembranous ossification pattern (Figure 1) [3,19–21]. Under this pattern, osteogenic cells derived from mesenchymal stem cells (MSCs) directly secrete osteoid and then mineralize as bone tissue [14]. Recent research has reported that capillary-like structures can be observed invading the avascular MSC layer prior to mineralization [21,22]. This indicates that ingrowth angiogenesis of the defect area is essential for craniofacial bone regeneration.

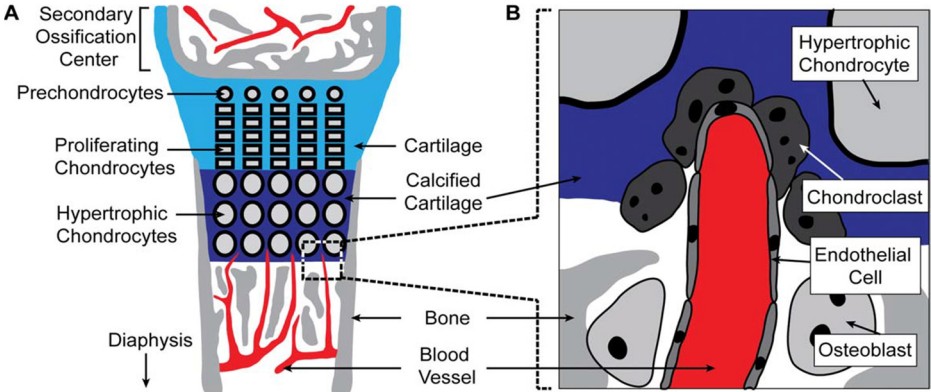

**Figure 1.** Schematic of the epiphyseal ossification of endochondral long bones, with emphasis on the process of capillary growth into calcified epiphyseal cartilage and subsequent trabecular ossification. (**A**) Chondrocytes differentiate from proliferating prechondrocytes within the growth plate. The chondrocytes are pushed toward the diaphysis by this continuous process and then enlarge under hypoxia, leading to mineralization of surrounding cartilage and the attraction of blood vessels required for bone formation. (**B**) Magnified view of bracketed zone from "A" showing capillaries, in association with chondroclasts, growing towards hypertrophic chondrocytes as a precursor to osteoblast activity and bone growth at the epiphysis. Reprinted with permission from Ref. [21], 2013, John Wiley and Sons.

However, there are still challenges in building a mature craniofacial neovascular system in the bone defect site. First of all, the defects of craniofacial bone, secondary to tumor resection or congenital craniofacial diseases, are usually critical-sized and the depth of defects is often thicker than 1 cm. The maximum cell–vessel distance to gain adequate oxygen supply and nutrition support is 200 μm [23,24], which means the transplanted or recruited cells may not survive, as minimal oxygen and nutrition can be obtained from host blood vessels, leading to the failure of new bone formation. Thus, how to establish

a well-organized and functional neovascular network within a short time to support cell proliferation and bone formation is the main challenge in craniofacial revascularization. Furthermore, the final functionalization of the neovascular vessel relies on the extent of anastomosis between neo and host vasculature [7]. Thus, therapeutic approaches for achieving successful anastomosis with resident vasculature is another issue that should be urgently addressed.

Furthermore, current studies demonstrate that blood vessels can be further classified as two different types. Type H vessels are characterized by high and positive expressions of CD31 and Endomucin. In contrast, type L vessels are characterized by low or negative expressions of CD31 and Endomucin. Type H vessels are reported to be located near the periosteum and endosteum of the diaphysis, while type L vessels are located in the bone marrow [25,26]. Evidence indicates that type H vessels can promote the proliferation and differentiation of osteoprogenitors and stimulate direct bone formation [27,28]. Therefore, how to increase the type H vessel ratio of neovascular network formation to promote craniofacial bone defect regeneration becomes another issue that should be further explored.

## 3. Various Vascularization Strategies in Craniofacial Bone Regeneration

Angiogenesis is a complex process involving extensive connections between vast growth factors, cells, and extracellular matrices (ECMs) [3,7,16–18]. After the bone defect occurs, the local hypoxic microenvironment and acute inflammation stimulate the release of pro-angiogenic growth factors from surrounding cells, which initiate the active proliferation of the endothelial cells (ECs) [8,29]. Afterwards, new blood vessels are formed, and then stabilized and remodeled by pericytes [30,31].

### 3.1. Cell Sources for Craniofacial Bone Vascularization

Vascularization involves vasculogenesis or angiogenesis; both these approaches rely highly on the functions of ECs and endothelial progenitor cells (EPCs) [32–35]. The craniofacial bone defect region is in need of oxygen and nutrients, and secretes proangiogenic molecules. As illustrated in Figure 2, ECs are triggered to be invasive (referred to as tip cells), lead the sprouts, and protrude filopodia [36]. Then, the protruded filopodia extend in response to the angiogenic signal source [36,37]. Tip cells are followed by stalk cells, which proliferate to elongate the sprout and form the fundamental vessel lumen [35,38,39]. Specification in migratory tip and proliferating stalk cells is dynamic, and ECs continuously compete for the lead position. Eventually, tip cells connect with surrounding tip cells from adjacent sprouts to form a new and stable vessel [35,39]. In addition, ECs have been proven to enhance the anastomosis between the neovessel system and host vasculature [7,40].

Considering that EC sources are limited in the craniofacial bone defect area, the EPCs, which are potentially derived from umbilical cord blood, peripheral blood, bone marrow, or human-induced pluripotent stem cells, are of great importance in vascular engineering because of their pluripotency and outstanding self-proliferation ability [41,42]. Studies have demonstrated that postnatal neovascularization is both directly and indirectly stimulated by EPCs [43,44]. Human EPCs have also been confirmed to be capable of forming a neovasculature in a critical-sized rat bone defect model, indicating that EPCs may be directly involved in the process of angiogenesis via differentiation into lumen-forming cells [45]. Furthermore, the expressions of proangiogenic vascular endothelial growth factors are higher in EPCs or EPCs/MSCs groups compared with those in MSCs alone, as well as the formation of blood vessels, confirming that EPCs are capable of initiating a host angiogenic response through cytokine secretion [44–46].

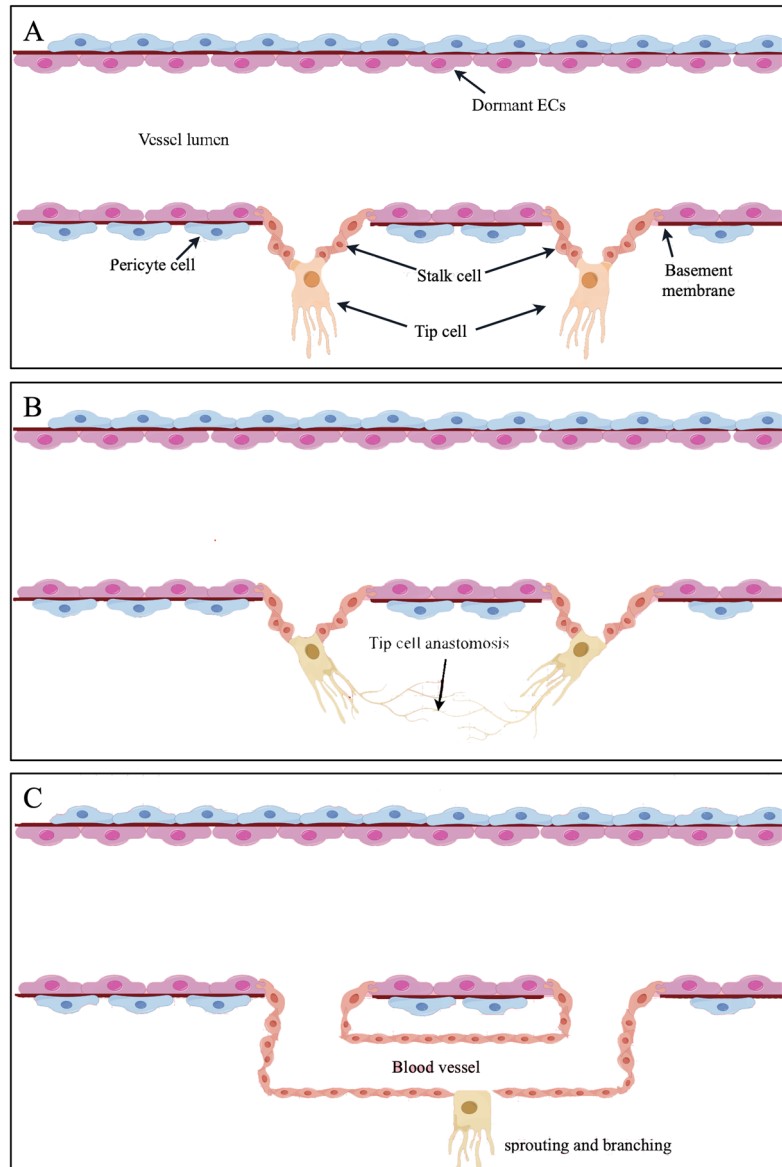

**Figure 2.** Illustrations of the sprouting and branching steps of blood vessels. (**A**) Sprout initiation and elongation. (**B**) Sprout anastomosis. (**C**) Vessel remodeling.

Except for ECs/EPCs, pericytes are the other composed cells of blood vessels (also referred to as perivascular cells, vascular smooth muscle cells, or mural cells). Pericytes play a crucial role in microvascular function, blood vessel stability, angiogenesis, and blood pressure regulation [30,31]. In addition, 22%~99% of the endothelium in capillaries is covered with pericytes, which are also found in pre-capillary arterioles and post-capillary venules. The high coverage rate seems to correlate positively with endothelial barrier properties. A larger coverage rate of pericyte leads to reduced EC turnover, whereas a lower coverage rate of pericyte results in enhanced proliferation and sprouting capacity of ECs. Furthermore, pericytes exhibit other important functions, such as contractile regulation of blood flow and the formation of ECM. Moreover, pericytes are embedded in the basement membrane of the vasculature and contact with surrounding ECs, resulting in an efficient communication, named pericyte–endothelial interactions, between the two cell types. Pericyte–endothelial interactions are necessary for the development and maintenance of a functional microcirculation in different tissues [47,48]. Currently, there are different views on the pericyte–endothelial interactions in different type of tissues. The most well-accepted view is that pericytes are recruited by stalk cells to support vessels. For example, research

in the retinal angiogenesis field shows that the retinal vascular network remains immature and is prone to rarefaction by ineffective stabilization until pericyte recruitment [49].

*3.2. Cell Signaling and Angiogenic Growth Factors*

The entire angiogenesis process consists of a series of growth factors and mediators of microenvironment components. The current knowledge of angiogenic biology has been widely expanding; one of the most significant factors is the vascular endothelial growth factor (VEGF) family. VEGF and its receptors (VEGFRs) play a prominent role in the activation of ECs in angiogenesis and osteogenesis [50–52]. ECs expressing high VEGFR2 signaling are called tip cells and promote the neighboring ECs' transfer as a stalk cell phenotype by upregulating the signaling of Notch ligand Delta-like 4 (DLL4). The transferring process is initiated by the activation of the NOTCH1 receptor of stalk cell via DLL4. The transferring in turn leads to the suppression of VEGFR2 and the concomitant induction of VEGFR1. The reciprocal regulation of VEGFR expression by Notch signaling reduces sensitivity to VEGF and thereby enforces stalk cell specification. The levels of VEGFRs, DLL4, and NOTCH1 are, however, constantly changing as ECs meet new neighbors. As a result, stalk cells can be relieved from tip cell inhibition and overtake the lead position, resulting in a dynamic position shuffle in the growing sprout [53,54]. The integrated regulation of VEGF and Notch is a prime example of a mechanism that allows ECs to sprout reiteratively in a concerted action, thereby ensuring robust network formation.

Platelet-derived growth factor-BB (PDGF-BB) is a member of the PDGF family, which is capable of improving the migration, proliferation, and differentiation of various mesenchymal cell types, such as EPCs and MSCs. ECs secrete PDGF-BB to recruit platelet-derived growth factor receptor–$\beta$ positive (PDGFR$\beta^+$) pericytes onto the neovasculature [55,56]. Su et al. demonstrated that bone angiogenesis was weakened when PDGF-BB was selectively knocked out in preosteoclasts. More bone angiogenesis was also obtained in *Pdgfb*-transgenic mice that overexpressed PDGF-BB (Figure 3) [55]. In addition to vessel formation, PDGF-BB–PDGFR signaling was reported to cooperate with DLL4–Notch signaling pathways to prevent excessive vascular sprouting and achieve a balanced and functional vessel network [57,58].

Hypoxia-inducible factor (HIF) is a transcription factor that alters the cell behavior in response to oxygen concentration and further affects angiogenesis [59,60]. HIF-1$\alpha$ is one of the most studied members of the HIF family. Studies have shown that HIF-1$\alpha$ is involved in angiogenesis or vascular remodeling processes through the so-called "HIF-1$\alpha$-VEGF axis". Under the stimulation of both angiopoietin-1 (Ang-1) and angiopoietin-2 (Ang-2), HIF-1$\alpha$ stimulates MSCs to secrete VEGF and inhibits the expression of the tissue inhibitor of metalloproteinase-3 (TIMP-3), an endogenous competitive inhibitor of the VEGF receptor (which mediates osteogenesis and angiogenesis) [61–64]. Furthermore, the ECs of Type H vessels have been reported to promote vascular growth via the HIF-1$\alpha$-VEGF axis and further communicate with perivascular osteoblasts through the Notch signal pathway for osteogenesis [26].

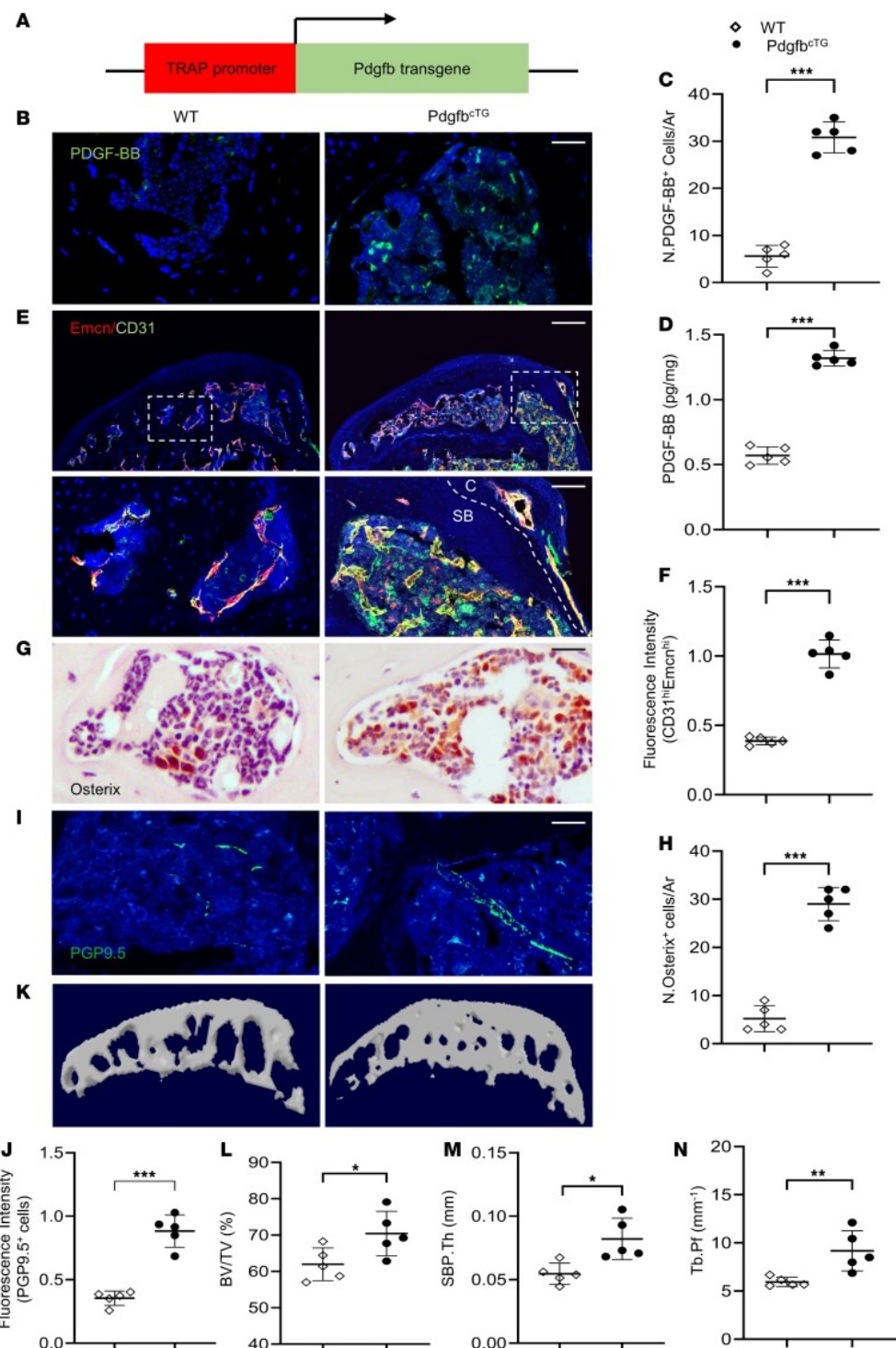

**Figure 3.** Transgenic mice expressing PDGF-BB in preosteoclasts recapitulate the pathological features of osteoarthritic joint subchondral bone. (**A**) Schematic diagram showing the TRACP5-*Pdgfb* transgene in transgenic mice (PdgfbcTG). (**B**–**N**) Knee joints were harvested from 5-month-old transgenic mice and WT mice. n = 5 mice per group. Immunofluorescence staining of PDGF-BB (green) and quantification of PDGF-BB$^+$ cells in tibial subchondral bone (**B**,**C**). Scale bar: 50 μm. *** $p < 0.001$. ELISA analysis of PDGF-BB concentration in tibial subchondral bone/bone marrow. *** $p < 0.001$ (**D**). Immunofluorescence staining of CD31 (green) and Emcn (red) with quantification of the intensity of CD31$^{hi}$ Emcn$^{hi}$ signal per tissue area in subchondral bone of the tibia (**E**,**F**). C—cartilage; SB—subchondral bone. Scale bars: 200 μm (**top**), 50 μm (**bottom**). *** $p < 0.001$. Immunohistochemical analysis of Osterix (brown) and quantification of Osterix$^+$ cells in tibial subchondral bone (**G**,**H**). Scale

bar: 50 μm. *** $p < 0.001$. Immunofluorescence staining of PGP9.5 (green) with quantification of the intensity of PGP9.5 signal per tissue area in subchondral bone of the tibia (**I,J**). Scale bar: 50 μm. *** $p < 0.001$. Three-dimensional μCT images (**K**) and quantitative analysis of structural parameters of subchondral bone: BV/TV (**L**), SBP. Th (mm$^{-1}$) (**M**) and Tb. Pf (mm$^{-1}$) (**N**). * $p < 0.05$, and ** $p < 0.01$. All data are shown as means ± standard deviations. Statistical significance was determined by unpaired 2-tailed Student's *t* test. Reprinted with permission from [55], 2020, American Society for Clinical Investigation.

In the context of biomaterials for craniofacial bone regeneration, the use of inorganic cations such as Ca$^{2+}$, magnesium (Mg$^{2+}$), and silicon (Si$^{4+}$) has gained attention due to their influence on mechanical and biological properties crucial for bone regeneration [18,65–68]. These cations do also play significant roles in neovascularization, which are crucial aspects of craniofacial bone regeneration [69–72]. Wang et al. displayed a sustained release of Mg$^{2+}$ from the piezoelectric Whitlockite scaffold and promoted angiogenic differentiation of BMSCs in vitro. Mg$^{2+}$ was further confirmed to remarkably form neobone with rich angiogenic expressions in an in vivo rat calvarial defect model (Figure 4) [73]. Liu et al. also reported the powerful angiogenic property of Mg$^{2+}$. In their study, MC3T3-E1 cells were treated with different concentrations of Mg$^{2+}$, and the secretion of PDGF-BB was promoted, which can effectively promote the angiogenic ability of HUVECs [74]. Wan et al. studied the synergistic effect of Mg$^{2+}$ and Si$^{4+}$. They fabricated hierarchical microspheres named PNM2, which can steadily release Mg$^{2+}$ and Si$^{4+}$ at an optimized ratio of 2:1 to match the process of vascularized bone regeneration at different stages. Then, a high volume and maturity of the vascularized neobone tissue was regenerated with PNM2 microspheres in a rat calvarial defect model [75]. Other cations, like Cu$^{2+}$ and Co$^{2+}$, also show proangiogenic activity [76,77]. Cu$^{2+}$ affects angiogenesis via regulation of the pERK1/2-foxm1-MMP2/9 axis [78]. Co$^{2+}$ has the capacity to stabilize HIF-1α and subsequently induce the production of VEGF, activating the angiogenic process by creating a hypoxia-mimicking condition [79].

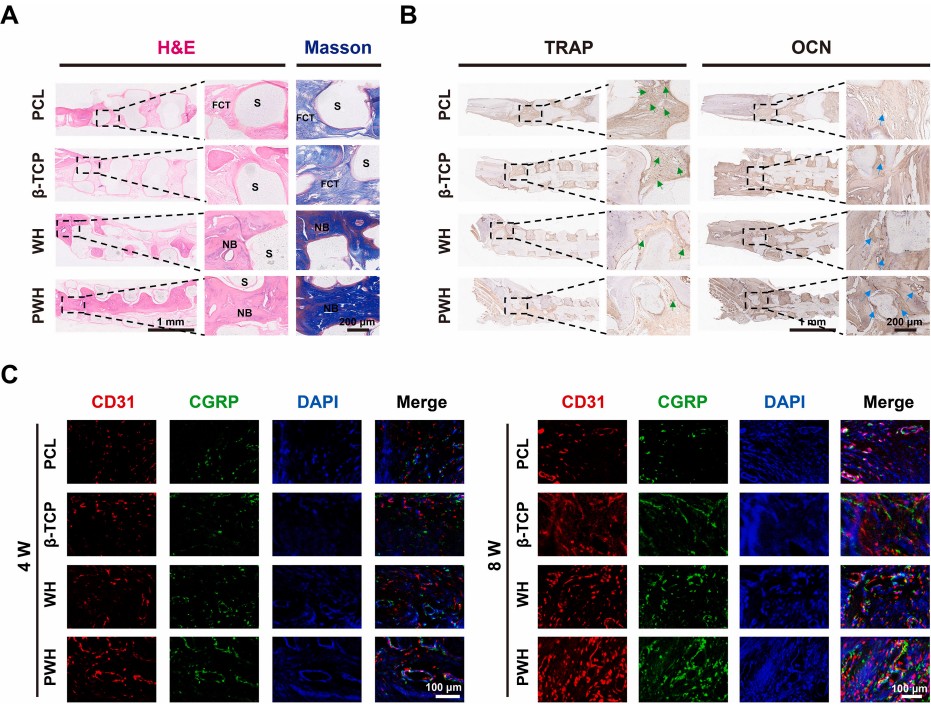

**Figure 4.** Histological, immunohistological, and immunofluorescence staining for the evaluation of bone regeneration. (**A**) H&E staining and Masson's trichrome staining for sections collected at 8 w

post-operation. (**B**) Immunohistochemical staining of TRAP and OCN expression at 8 w post-operation. (**C**) Immunofluorescence staining of CD31 (red), CGRP (green), and DAPI (blue) expression at 4 w and 8 w post-operation. FCT: fibrous connective tissue. NB: newly formed bone tissue. S: spaces in scaffolds. Reprint from [73].

A breakthrough in the additional application of miRNA in 2006 moved angiogenesis to another level [80]. Zhuang et al. investigated the angiogenic effect of miR-210-3p. They found miR-210-3p can hinder EFNA3 expression and subsequently activate the PI3K/AKT pathway, enhancing the proliferation, migration, and angiogenesis of ECs [81]. Castaño et al. also reported the synergetic effect of dual delivery of two miRNAs. MiR-210 mimics and miR-16 inhibitors were released from a collagen–nanohydroxyapatite scaffold system to enhance angiogenesis and osteogenesis, resulting in accelerated rat calvarial bone defect repair [82]. There are also many other angiogenic miRNAs, such as miR-378, miR-126, and Let-7, that target various signaling molecules. These miRNAs target various aspects of angiogenesis via endothelial cell function, blood vessel formation, and related growth factor signaling (Table 1).

**Table 1.** Major angiogenesis pathways and potential therapeutic miRNAs.

| Target Pathway | miRNA | Target Signaling Pathway |
|---|---|---|
| Activation of ECs | miR-210-3p | EFNA3/PI3K/AKT [81,83] |
| | miR-378 | Sufu [84] <br> Fus1 [85] |
| | miR-126 | SPRED1/Ras/Erk [86,87] <br> PIK3R2 [88] <br> VCAM-1 [89] |
| Sprouting, migration, and tubulogenesis of ECs | miR-17-92 | ERK/ELK1 [90] |
| | Let-7f-5p | DUSP1/Erk1/2 [91] |

*3.3. Co-Culture Systems with Different Cell Types or Growth Factors*

Co-culture strategies involving different cell types and growth factors have indeed found application in the field of craniofacial bone vascularization. These approaches leverage the interactions between various cell types and signals to enhance osteogenesis and angiogenesis. According to previous studies, the co-culture system may strengthen the participating cells isolated from different tissues, showing enhanced cell functions. Co-transplantation of EPCs and osteogenic stem cells has been widely accepted in vascularized bone regeneration due to their respective angiogenic or osteogenic potentials reported in different studies [92–94]. Cells harvested from bone marrow aspirates possess osteogenic ability and can also be induced to form tubelike structures and ECs. Studies have shown that EPCs or MSCs can secrete bone morphogenetic protein–2 (BMP-2), a potent inducer of osteogenesis [95]. In turn, MSCs have secreted angiogenic cytokine to promote the migration of EPCs via PDGF-BB and vessel formation [55,56]. Except for different cells, growth factors are also dually applied within the same system. PDGF-BB co-expression with VEGF can prevent the VEGF-related aberrant angiogenesis. Within the system, the VEGF–VEGFR2 induces vascular sprouting and the PDGF-BB–PDGFRβ system can synchronize with DLL4–Notch signaling to prevent excessive vascular sprouting at the same time, preventing imbalanced expression or activation of each of these signaling components and vascular dysfunctions [96].

*3.4. Biological Requirements for Biomimetic Scaffolds Used for Craniofacial Bone Vascularization*

Scaffold materials used in the field of craniofacial regeneration serve a critical role beyond simply providing a structural framework; they must also support vascular regeneration in addition to promoting osteo-induction and osteo-conduction (Table 2).

**Table 2.** In vivo biomaterial scaffold research about vascularization and craniofacial bone regeneration during the last 5 years.

| Author and Year | Biomaterial Scaffolds | Bioactive Agent | Implanted Cells | Animal Model | Observation Period | Osteogenesis | Angiogenesis |
|---|---|---|---|---|---|---|---|
| Yaxi Sun, Dent Mater, 2023 [97] | Calcium phosphate cement scaffold (CPC) | Metformin | hPDLSCs | Critical-sized defect of rat cranium | 12 weeks | 9 folds by control | 3 folds by control |
| Ruochen Luo, Biomed Mater, 2021 [98] | Poly(lactide-co-glycolide) microspheres | $Mg^{2+}$ and $La^{3+}$ | --- | Critical-sized defect of rat cranium | 8 weeks | Enhanced | Enhanced |
| Nurul Aisyah Rizky Putranti Cells, 2022 [99] | Carbonate hydroxyapatite (CAP) granules | BMP-2 | SHED | Critical-sized defect of immunodeficient mice cranium | 12 weeks | Enhanced | Enhanced |
| Kun Liu, Regen Biomater, 2020 [100] | Mineralized collagen | BMP-2 and VEGF | --- | Mandibular defects of rabbits | 12 weeks | Enhanced | Enhanced |
| [101] | GM/Ac-CD/rGO hydrogel | --- | --- | Critical-sized defect of rat and mice cranium | 8 weeks | Enhanced | Promotes type H vessel formation |
| Omar Omar, Proc Natl Acad Sci U S A, 2020 [102] | Bioceramic (biocer) implants | --- | --- | Skull defect of ovine | 12 months | Enhanced | Enhanced |
| Yaohui Tang, Theranostics, 2020 [103] | Injectable gelatin-based µRB hydrogel | BMP-2 | ASC | Critical-sized defect of immunodeficient mice cranium | 8 weeks | Enhanced | --- |
| Yuanjia He, Stem Cell Res Ther, 2020 [104] | HA/Col scaffold | --- | EPCs and ASC | Critical-sized defect of rat cranium | 8 weeks | Enhanced | Enhanced |
| Maxime M Wang, Sci Rep, 2019 [105] | 3D-printed bioceramic scaffolds | Dipyridamole | --- | Unilateral alveolar defect of rabbits | 24 weeks | Enhanced | --- |
| Weibo Zhang, Front Bioeng Biotechnol, 2020 [106] | E1001(1K)/β-TCP scaffolds | Tyrosine-derived polycarbonate | hDPSCs and HUVECs | Mandible defect of rabbits | 3 months | Enhanced | Enhanced |
| Marley J Dewey, Biofabrication, 2021 [107] | Mineralized collagen/PCL composites | --- | --- | Critical-sized defect of porcine ramus | 10 months | Enhanced | Enhanced |

**Table 2.** *Cont.*

| Author and Year | Biomaterial Scaffolds | Bioactive Agent | Implanted Cells | Animal Model | Observation Period | Osteogenesis | Angiogenesis |
|---|---|---|---|---|---|---|---|
| Qian-Qian Wan, ACS Appl Mater Interfaces, 2022 [108] | Eggshell membranes | Cerium oxide | --- | Critical-sized defect of mice cranium | 8 weeks | Enhanced | Enhanced |
| Yue Kang, Biofabrication, 2023 [109] | Hybrid scaffolds | Exos isolated from hASC | --- | Critical-sized defect of immunodeficient mice cranium | 10 weeks | Enhanced | Enhanced |
| Zeqing Zhao, J Dent, 2023 [110] | Calcium phosphate cement (CPC) scaffolds | Human platelet lysate | hPDLSCs and hUVECs | Critical-sized defect of immunodeficient mice cranium | 12 weeks | 4 folds by control | 7.9 folds by control |
| H Autefage, Biomaterials, 2019 [111] | Bioactive glass-based scaffold | Strontium | --- | Femoral condyle defect of ovine | 12 weeks | Enhanced | --- |
| Tania Saskianti, Clin Cosmet Investig Dent, 2022 [112] | Hydroxyapatite | --- | SHED | Mandibular defect of rats | | Downregulation of MMP-8 | Upregulation VEGF expressions |
| W Ma, J Dent Res, 2021 [113] | Col scaffold | Galanin | --- | Periodontitis-treated mice | 6 weeks | Enhanced | --- |
| Tsuyoshi Kurobane, Acta Biomater, 2019 [114] | Octacalcium phosphate/gelatin composite (OCP/Gel) | --- | --- | Critical-sized defect of immunodeficient mice cranium | 4 weeks | --- | Enhanced |
| Mirali Pandya, Int J Mol Sci, 2021 [115] | Collagen/erythropoietin (EPO) scaffold | EPO | | First maxillary molars extracted rats | 8 weeks | Enhanced | enhanced |
| TaichiTenkumo, Regen Ther, 2023 [116] | A triple-functionalized paste of CAP | DNA and siRNA | --- | Femoral head defect of rats | 21 days | Enhanced | --- |
| Hyeree Park, Mater Sci Eng C Mater Biol Appl, 2021 [117] | DC-S53P4 bioactive glass hybrid gels | --- | DPSCs | Critical-sized defect of immunodeficient mice cranium | 8 weeks | Enhanced | Enhanced |

### 3.4.1. Surface Morphology

The surface characters of scaffolds are closely related to the cell adhesion, proliferation, and function of blood vessel-forming related cells, which could eventually affect the formation of the functional vessel network. Surface modification includes chemical modification and physical modification, and methods of surface modification usually include immersion [118], coating [119–123], and plasma treatment [124,125]. Porous polyetheretherketone

(PEEK) scaffolds are modified via polydopamine and $Mg^{2+}$ physically deposited on the surface. After surface modification, the hydrophilicity of PEEK scaffolds is significantly enhanced, and bioactive $Mg^{2+}$ could be released, contributing to the reinforced formation of osteogenic H type vessels in a rabbit femoral condyle model [15].

Except for the physical modifications, chemical modifications of the scaffold surface can also facilitate the adhesion and biological behaviors of blood vessel-forming related cells. Among various studies, several amino acid sequences, such as Arg–Gly–Asp (RGD) [126,127], have triggered intensive studies for the enhancement of EC adhesion by the establishment of a ligand-modified surface and established capillary structure. Apart from the well-known amino acids, Hao et al. successfully identified the αvβ3 integrin ligand LXW7 with the help of unnatural amino acids [128]. The following research found that LXW7 showed a stronger binding affinity to primary EPCs/ECs. In addition, an LXW7-treated surface exhibited proliferation, migration, and tubule formation through increased VEGFR2 phosphorylation [129].

### 3.4.2. Porous Characters

A porous scaffold is the prerequisite of cell ingrowth, oxygen supply, and nutrient transport. Studies have illustrated that different properties of scaffold pores can affect cell type and behavior. The suggested range of diameters is considered from 200 to 350 μm for bone regeneration [130], while the favored pore size for revascularization is larger than 400 μm [7]. Moreover, large pore sizes have been reported to be beneficial for cell viability but harmful for cell seeding. The above contradiction reminds us that a monomodal scaffold is out of date. A scaffold with multiple pore size is urgently needed in order to fulfill the best osteogenesis and angiogenesis at the same time.

### 3.4.3. Angiogenic GF Release Property

Sufficient oxygen, enough nutrients, and a large amount of different types of regenerative cells can be supplied to the defect area, allowing the inflammation factors, metabolic wastes, and necrotic tissue to be removed in time in the presence of the dense vascular network. Angiogenic growth factors, including VEGF, have recently attracted much more attention [131]. However, the narrow range of therapeutic windows of VEGF limits clinical promotion. It has been reported that the concentrations of VEGF determine the fate of the tissue regeneration process. Higher concentrations of VEGF are shown to result in unfavorable effects, such as increased permeability and being prone to create malformed and non-functional blood vessels [131,132]. Furthermore, a large proportion of VEGF degraded rapidly because of the short half-life before coming into effect when being released into the biological milieu [131]. Therefore, it is necessary to achieve a spatial and temporal release of VEGF to prolong its activity. Wernike et al. reported that, when VEGF was introduced in the cranial defects of mice and was slowly released via implanted biomimetic BCP ceramics, more dense vessels and more regular vessel morphology could be visualized with intravital microscopy compared with burst-released VEGF [132]. Burger et al. also reported that over-expression of VEGF was associated with paradoxical bone loss. They controlled the distribution of the VEGF dose with factor-decorated matrices and observed both improved vascularization and bone formation in orthotopic critical-size defects compared with burst-released VEGF [133].

### 3.5. Scaffold-Free Technique

Although biomimetic scaffolds provide the supportive structures for cell ingrowth, ECM depositions, and tissue regeneration, they still bear a non-negligible drawback, i.e., incomplete biodegradability [134]. This may induce chronic inflammation and hinder the complete regeneration of bone defect with neotissue. Under this circumstance, scaffold-free tissue engineering, also referred to as cell sheet engineering (CSE), has been developed in recent years. In scaffold-free tissue engineering, cells are directly assembled or aggregated to form a tissue-like structure. This approach relies on the temperature-responsive cell culture

technology and inherent ability of cells to self-organize and interact with one another to create functional tissue structures (Figure 5). Cells are cultured on temperature-responsive culture dishes or surfaces coated with a temperature-responsive polymer, such as poly(N-isopropylacrylamide) (PIPAAm), to form monolayer sheets. These polymers change their properties with temperature, allowing cells to adhere to the surface at 32 °C and detach as a sheet at 37 °C. Thus, the cell–cell junctions can be preserved, avoiding ECM damage caused by proteolytic enzymes [134]. Since the thickness limitation of 3D constructs without vascular networks is no more than 80 µm [135], co-culture cell sheet approaches have been proposed. Human umbilical vein endothelial cells have been reported to be co-cultured within human myoblast sheets to form capillary-like structures within the construct. Then, increased neovascularization and graft survival after transplantation were obtained [136].

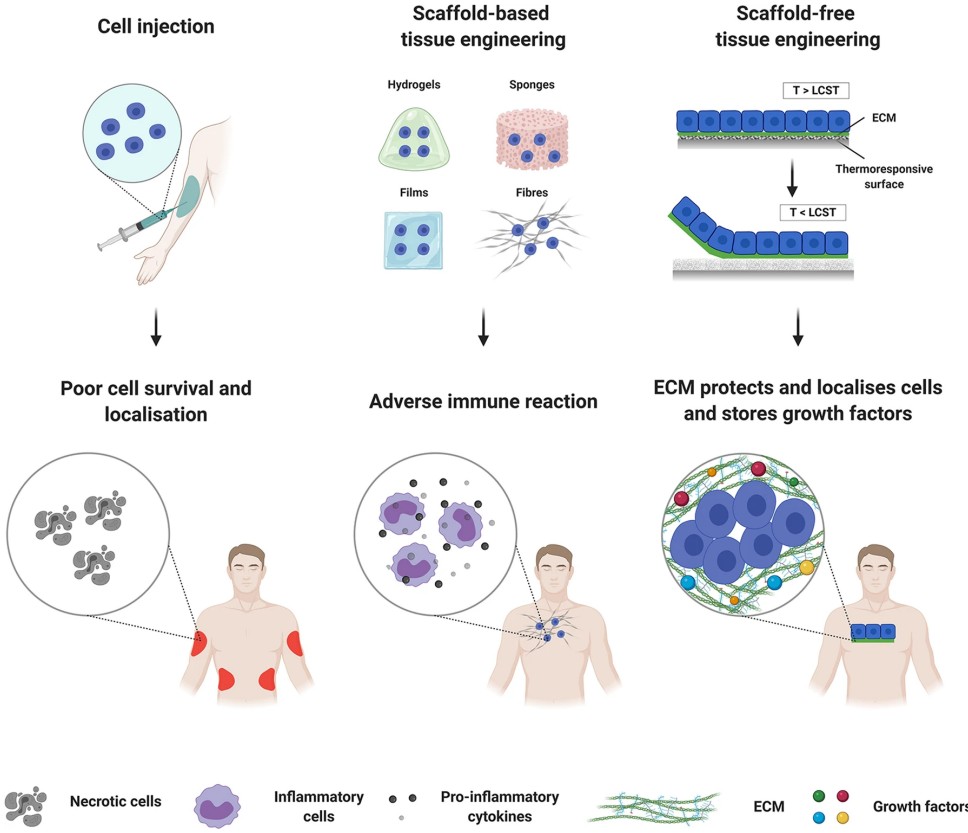

**Figure 5.** Cell-based tissue engineering therapies. Reprint from [134].

## 4. Conclusions and Further Expectations

Vascularization in craniofacial bone tissue engineering is a critical aspect of regenerating bone tissue in the craniofacial region. Craniofacial bone defects can result from various causes, including trauma, congenital anomalies, or surgical resection due to disease. To successfully regenerate bone in this region, ensuring the development of a functional vascular network is essential. This review paper therefore focused on the use of angiogenic growth factors, cell-based vascularization strategies, and surface morphology, porous characters, and the angiogenic release property of scaffolds.

Although experiments on neovascularization have shown encouraging results, how to establish the functional neovascular vessel with fast and mature anastomosis between neo and host vasculature and how to increase the osteogenic H type vessel ratio remains challenging. Further studies are required to address these issues and explore angiogenic mechanisms in craniofacial bone reconstruction.

**Author Contributions:** Conceptualization and supervision, H.Q. and J.C.; writing—original draft preparation, J.C.; writing—review and editing, Y.D., J.Y. and G.Z. All authors have read and agreed to the published version of the manuscript.

**Funding:** This research received no external funding.

**Conflicts of Interest:** The author declares no conflicts of interest.

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
