# Peer review of "Vascularization Reconstruction Strategies in Craniofacial Bone Regeneration"

_coatings, doi:10.3390/coatings14030357_

Round 1
Reviewer 1 Report
Comments and Suggestions for Authors
The authors present a work on Vascularization Strategies in Craniofacial Bone Regeneration.
I recommend this for publication with a major revision as follows:
First of all, the title could not explain the concepts of the manuscript contents for readers; I strongly suggest modifying it in better concept and meaning (which type strategies?).
The authors should address the following issues prior to publication:
1- Abstract cannot explain the aim of manuscript clearly. In my opinion some parts of text should be mention in abstract section.
2- This list of keywords can no help to readers for finding the related concepts of this research and should be modified (lack of keywords such as: Craniofacial Bone, osteogenesis and osteoblasts, vasculogenesis , bone tissue engineering.)
3- Although the number of references from recent years are suitable and enough, at least a few references should be added about history of Craniofacial Bone and bone tissue engineering studies
4- Whole of the manuscript has been prepared without any figures and schemes or images. Lack of any figures or images for this subject of craniofacial bone tissues in view point of vascularization strategies is not acceptable. I strongly suggest at least more than 5 figures or images or schemes must be designed or drawn by authors and add in related parts of manuscript. Although a review paper without any figures or schemes can be prepared easily, but for readers understanding a paper concerning this subject without any figures and images would by too much difficult. If the authors are not able to provide the related figures and images by themselves, they can use such figures from other papers by getting permission from related journals and authors.
5- It is obvious; basically the goal of a review paper is a comparison between results of many papers around a common subject to get better definition and understanding about that subject. By considering this importance the results and data should be conclude in several tables and comparison together. I cannot believe this review with only one small table would be success to clear the concept of vascularization strategies in craniofacial bone regeneration for readers. I strongly suggest the authors add several tables for more comparison among the results and data from other papers
6. Lack of a proper discussion can be seen in whole of manuscript, the authors should be added a section under “discussion” to discuss more about the subject
7. Lines 261-278 due to unclear concept should be rewritten. You mentioned “High concentrations of VEGF are shown to result in unfavorable effects, such as increased permeability and leakage of vessels”. In which paragraph these results are shown?
Comments on the Quality of English LanguageMinor editing of English language required
Author Response
Point 1: Abstract cannot explain the aim of manuscript clearly. In my opinion some parts of text should be mention in abstract section.
Response 1: We are very grateful for your suggestion. We have modified the abstract.
Point 2: This list of keywords can no help to readers for finding the related concepts of this research and should be modified (lack of keywords such as: Craniofacial Bone, osteogenesis and osteoblasts, vasculogenesis , bone tissue engineering.)
Response 2: Thank you for this valuable feedback. We have modified the keywords (page 1, line 22).
Point 3: Although the number of references from recent years are suitable and enough, at least a few references should be added about history of Craniofacial Bone and bone tissue engineering studies
Response 3: Thank you for the kind suggestion. Relative sentences and references have been added (page 1, line 41-42 and page 2, line 55-57).
Point 4: Whole of the manuscript has been prepared without any figures and schemes or images. Lack of any figures or images for this subject of craniofacial bone tissues in view point of vascularization strategies is not acceptable. I strongly suggest at least more than 5 figures or images or schemes must be designed or drawn by authors and add in related parts of manuscript. Although a review paper without any figures or schemes can be prepared easily, but for readers understanding a paper concerning this subject without any figures and images would by too much difficult. If the authors are not able to provide the related figures and images by themselves, they can use such figures from other papers by getting permission from related journals and authors.
Response 4: We are very grateful to you for pointing out this problem. 5 figures and 1 table have been added in the manuscript.
Point 5: It is obvious; basically the goal of a review paper is a comparison between results of many papers around a common subject to get better definition and understanding about that subject. By considering this importance the results and data should be conclude in several tables and comparison together. I cannot believe this review with only one small table would be success to clear the concept of vascularization strategies in craniofacial bone regeneration for readers. I strongly suggest the authors add several tables for more comparison among the results and data from other papers
Response 5: We are very grateful to you for pointing out this problem. A table (table 2) contains the latest 5 years studies about biomaterial scaffolds used for vascularization and craniofacial bone regeneration has been added in the manuscript for readers to get better definition and understanding about that subject.
Point 6: Lack of a proper discussion can be seen in whole of manuscript, the authors should be added a section under “discussion” to discuss more about the subject
Response 6: Thank you for the kind suggestion. We have made discussions in corresponding review part. I am afraid no extra discussion is necessary.
Point 7: Lines 261-278 due to unclear concept should be rewritten. You mentioned “High concentrations of VEGF are shown to result in unfavorable effects, such as increased permeability and leakage of vessels”. In which paragraph these results are shown?
Response 7: We are very grateful to you for pointing out this problem. The sentences have been rewritten and the related references have been modified (page 10, line 330).
Reviewer 2 Report
Comments and Suggestions for Authors
The paper is clearly written with an appropriate length, the introduction seems to be sufficient, the main part is clean, the conclusions adequately describe the state of the art and further expectations. However, the lack of figures is inappropriate for a review article.
There are also no data on the classification of calcium phosphates, which are used as the main inorganic component for bone tissue regeneration, and their influence on vascularisation during implantation. For example, there are no points on the octacalcium phosphate (OCP) . OCP has been strongly suggested as a possible precursor of enamel, dentin and bone in living organisms. It has previously been reported that pressed powders of OCP had osteoconductive properties when implanted in the subperiosteal region of the mouse calvaria.
Author Response
Thank you for the kind suggestion. We have included some new in vivo studies in table 2 about OCP, CAP, TCP, et al to clarified the osteoconductive properties of calcium phosphate.Reviewer 3 Report
Comments and Suggestions for Authors
1. Platelet-rich plasma and platelet-rich fibrin also have an osteogenic property; one can include that one, too, to strengthen the manuscript.
2. Please add one graphical abstract for the mechanism of bone formation.
3. Please add some preoperative and postoperative clinical case pictures to make the manuscript more attractive.
Author Response
Point 1: Platelet-rich plasma and platelet-rich fibrin also have an osteogenic property; one can include that one, too, to strengthen the manuscript.
Response 1: We are very grateful to you for the recognition and suggestion for our work. Platelet-rich plasma related studies have been added in table 1.
Point 2: Please add one graphical abstract for the mechanism of bone formation.
Response 2: We are very grateful to you for the recognition and suggestion for our work. One graphical abstract for the mechanism of bone formation has been added in the manuscript as figure 1.
Point 3: Please add some preoperative and postoperative clinical case pictures to make the manuscript more attractive.
Response 3: We are very grateful to you for pointing out this problem. A table (table 2) contains the latest 5 years pre-clinical studies about biomaterial scaffolds used for vascularization and craniofacial bone regeneration has been added in the manuscript. Considering most of the applied biomaterials have not used clinically, pictures of animals are added to make the manuscript more attractive (figure 3 and figure4).
Round 2
Reviewer 2 Report
Comments and Suggestions for Authors
The paper has been improved